*Correspondence*

# Reply to "CRISPR screens are feasible in *TP53* wild-type cells"

Emma Haapaniemi[1], Sandeep Botla[1] iD, Jenna Persson[1,2], Bernhard Schmierer[1,2] iD & Jussi Taipale[1,3,4] iD

Reply to: **KR Brown *et al*** (August 2019)

Our paper "CRISPR/Cas9 genome editing induces a p53-mediated DNA damage response" (Haapaniemi *et al*, 2018) in *Nature Medicine* demonstrated that the efficiency of precision gene editing by homologous recombination is impaired in the presence of functional p53. We provided evidence that Cas9 nuclease activity causes a p53-dependent, transient G1-phase arrest, which we hypothesized to cause the observed p53-dependent decrease in precise, template-mediated gene editing. Based on these results, we advised caution in therapeutic gene editing, where selecting cells that edit well could lead to inadvertent co-selection of cells with a suboptimal DNA damage checkpoint. Selection (or use) of such cells should be avoided in a clinical setting, as they may be at an increased risk for malignant transformation. In a co-submitted manuscript, Ihry *et al* (2018) reported similar findings in another cell line. These results have since been independently reproduced by several groups using RPE-1 cells and other normal human cell types (Li *et al*, 2018; Cullot *et al*, 2019; preprint: Geisinger & Stearns, 2019; Schiroli *et al*, 2019; preprint: van den Berg *et al*, 2019). The significant attention that our work received was due to the relevance of our conclusions for genome editing in normal human cells and for the development of cell-based therapies. Many investigators have since taken a constructive approach to the challenge posed by p53 activation in their respective systems and have taken major steps in designing reagents and protocols that decrease the p53 response in clinically relevant cells (Schiroli *et al*, 2019).

The correspondence by Brown *et al* creates the impression that the main topic of our paper was CRISPR loss-of-function screening and that we had argued that such screening is not feasible in p53 wild-type cells. This is incorrect. We had originally discovered the p53 effect when comparing the CRISPR knock-out screening performance of several Cas9 expressing tumor cell lines and the normal, hTERT immortalized human cell line RPE-1. To demonstrate the motivation for our study, we included our initial genetic screening experiment in RPE-1 cells. This screen identified the p53/p21/pRB pathway, but did not show the clear "drop-out" of guides against essential genes that we had observed in a set of tumor cell lines. Follow-up experiments indicated that the drop-out performance of a p53 null RPE-1 line was superior to that of the wild-type, which provided a first clue that Cas9 might induce a p53-mediated DNA damage response. We then continued to analyze the effect of p53 down-regulation on precision genome editing, which was the actual topic of our paper.

Due to our main focus, the medium of publication, and the very brief format, we did not extensively discuss the implications of our work on applications of Cas9 in basic research in general and in CRISPR "drop-out" screening in particular, only adding two short sentences within the main text: "*DSBs introduced by CRISPR–Cas9 trigger a transient, p53-dependent cell cycle arrest mediated through p21 and pRB, irrespective of the locus targeted. This generic penalty of DNA cutting masks guide-specific effects, hampering guide dropout screens that are aimed at identifying genes whose loss leads to cell death or decreased cell proliferation.*"

In their correspondence, Brown *et al* first misinterpret and then object to this statement, and put forward two main lines of argument: (i) CRISPR drop-out screens are feasible in p53WT cells and the induction of a p53 response is "not a major concern", and (ii) screening should always be performed in a specific way that includes pre-selection of cell lines or clones that perform exceptionally well in screening.

Our statement about drop-out screening being "hampered", i.e., being made more difficult, referred to p53WT RPE-1 hTERT cells only, and not to tumor cells, many of which display no p53-mediated DNA damage response, or a weaker response than that observed in RPE1 cells (preprint: Geisinger & Stearns, 2019). We agree with Brown *et al* that drop-out screening using Cas9 nuclease can be feasible in p53WT cells under conditions that are different from those used in our work; this is in fact what our original transient arrest model (above in italic typeface) predicts, and we have never claimed otherwise.

What our data do show is increased noise caused by the DNA damage checkpoint in RPE-1 p53WT cells. This result was obtained in a direct side-by-side comparison of screening performance of RPE-1 WT and RPE-1 p53$^{-/-}$ cells using the same virus library—an experiment which Brown *et al* notably have not performed. Although definite conclusions cannot be made based on

1  Karolinska Institutet, Stockholm, Sweden
2  Science for Life Laboratory, Stockholm, Sweden
3  University of Helsinki, Helsinki, Finland
4  University of Cambridge, Cambridge, UK. E-mail: ajt208@cam.ac.uk
**DOI** 10.15252/msb.20199059 | Mol Syst Biol. (2019) 15: e9059

the evidence they provide due to this lack of paired data with the same library, the performance of the Brown et al p53 null and WT screens is in fact consistent with the p53 response also hampering their screens: In a MAGeCK analysis (Li et al, 2014) of Brown et al's raw data, the p53 null screen recovers approx. three times as many hits as the p53 WT screen at an FDR < 0.05 (533 vs. 170, average of two replicates), despite the fact that the p53 null screen was performed using inferior experimental design and a less advanced library (see below). We believe that the difference between our analysis with standard statistical tools and Brown et al's complex machine-learning measure is due to an unfair comparison between RPE-1 p53 null and wild type. In their analyses, one out of three WT experiments ("UBC", using TKOv3) outperforms their single p53 null experiment (done with TKOv2), likely because (i) it uses more partial replicates, (ii) the experimental design is more advanced, using a bottleneck after transduction to decrease clonal variance (Michlits et al, 2017; Schmierer et al, 2017), (iii) it uses better guide-RNA library, and (iv) the measure used selectively exaggerates its performance due to overfitting (the guides for TKOv3 but not TKOv2 were designed based on the essential gene set they use to test performance). In contrast, our simpler analysis using the same number of replicates for null and wild-type and eliminating potential for overfitting leads to a result that is converse to that reported by Brown et al, and consistent with the model presented in our original publication.

In addition to not having performed the same side-by-side experiment using a fair measure of performance, the design principles of Brown et al's screens are materially different from ours: Our screen was designed to be robust to clonal variation and thus allows more sensitive comparison of assay performance between cell types (Brown et al panel E). The design by Brown et al seeks to decrease variation between cells in order to optimize detection of drop-outs in all cell types (panel B). Their process involves separate and careful optimization of screening performance for each cell line, making comparisons of screening performance between cell types very difficult, if not impossible. Both design principles are compromises, and as a consequence, the screens do not perform equally well in detecting the converse features. Brown et al

accept that p53 loss is selected for, but argue that p53 activation does not have major effects on the performance of CRISPR knock-out screens in their analyses. We believe that their failure to detect the p53 effect is in large part because, to optimize screening performance, they selected clonal cell lines that edit efficiently or "screen well", used excessive normalization procedures, and focused on thresholded calls and (overfit) measures that are too lenient to detect the underlying differences in performance. Under such synthetic tests, the true extent of the variance caused by p53 is normalized away.

Thus, consistent with a p53 response hampering screens in normal (untransformed) human cells, MAGeCK analysis of the raw data reveals that under both our and Brown et al's screen designs the performance of the p53 null RPE-1 is higher than that of RPE-1 p53 WT. If one makes the unsafe assumption that the clonal cell lines used in Brown et al are fully representative of parental RPE-1 cells, the observation that the overall drop-out performance in their screens is higher is explained most parsimoniously by the model we proposed in Haapaniemi et al. The key variable is the rate of Cas9-induced mutagenesis relative to the cell cycle time. If the mutation rate is much faster than the cell cycle time, all cells will be edited within one cell cycle, and p53WT cells will arrest temporarily during the same cycle. In this case, the p53-dependent increase in noise is expected to be at its minimum and can be missed by investigators if only highly processed data are analyzed, or if single cell clones of different cell lines are compared with each other. If cutting is slower than the cell cycle time, there is little difference for a p53 null cell. For a p53 wild-type cell, however, division of cells that contain DSBs will be slowed by p53 (preprint: Geisinger & Stearns, 2019), whereas cells that happen to edit poorly will divide. This will still allow efficient detection of enriched guides, but lead to a runaway loss of precision in the drop-out part of the screen. The RPE1-Cas9 cells Brown et al use seem to heavily overexpress Cas9 (Zimmermann et al, 2018). This is expected to lead to very fast kinetics of DNA cutting, and more persistent binding of Cas9 to its target site after the cut (preprint: Geisinger & Stearns, 2019). Thus, the very high Cas9 expression in their cells (Zimmermann et al, 2018) is a plausible explanation

for why both screens detected the positive selection for loss of p53, but only our screens clearly distinguished between the drop-out performance of p53 wild-type and p53 null cells. We would like to note that it is our discovery of differences in the responses of normal and tumor cells to genome editing that attracted attention to our work, not the failure to find drop-outs.

We wish to reiterate that selection of cells that perform well in CRISPR/Cas9 experiments is not without risk. Cas9 cutting introduces a low level of a very genotoxic form of DNA damage, induces a DNA damage response, and can lead to severe chromosomal aberrations or large deletions (Kosicki et al, 2018; Cullot et al, 2019); at least in primary human cells, the frequency of occurrence of the large deletions is p53 dependent (Cullot et al, 2019). The DNA damage response varies as a function of experimental conditions and cell types and can induce cell cycle arrest or cell death. As p53 integrates many cellular stresses, the magnitude of the response will also depend on the targeted site (preprint: van den Berg et al, 2019), the delivery method and other experimental design parameters. The measured variables (e.g., fitness, cell cycle) will interact differentially with DNA damage and p53. Irrespective of the screen, all hit genes must thus be formally considered synthetic with the DNA damage response, which is a confounding variable in CRISPR nuclease-based screens. The effect can be mitigated to some extent in screens targeting single coding regions as highly specific guides can be selected; much larger amount of DNA damage is expected to occur in screens where less optimal guides are needed (e.g., those targeting non-coding regions), or where multiple genes are targeted simultaneously.

Selection of cells that perform well in CRISPR screens can help to identify clones with fast editing kinetics, partially mitigate the effect of p53, and increase technical precision. However, it can also easily lead to selection of cells that are unrepresentative of the whole population, or inadvertent selection of cells that have qualitative or quantitative defects in sensing or repairing DNA damage—potential issues that Brown et al have not addressed. Selection of clones is particularly problematic when analyzing genetic interactions or studying highly aneuploid or genetically unstable cancer cells. The biologically most relevant results are

obtained by using populations of cells, optimally in combination with lineage tracing by UMIs to sample the range of cellular heterogeneity (Michlits *et al*, 2017; Schmierer *et al*, 2017). We realize that in some experimental settings, clones are still preferable; in such case, we would advocate careful genetic analysis of the clones (e.g., karyotyping), and systematic labeling of clonal lines in publications to avoid confusion.

Even though we agree with most of the advice given by Brown *et al* on how to design CRISPR screens to optimize technical performance in a clone derived from a given cell line, we disagree with the broader implication that a particular type of experiment must always be performed in a particular way. Design of experiments must be based on the scientific question addressed and cannot blindly follow rules designed for other purposes. It is admittedly counterintuitive that even changes to experimental design that seem universally beneficial and completely harmless—such as optimizing technical performance within a given framework—can sometimes hamper scientific discovery. Imposing blanket "quality control standards" and dismissing experiments that apparently fall short of them can lead to failure to detect variables that affect "quality" (Brown *et al* panels A–D) and to failure to ask the question: Why does experiment A have lower "quality" than experiment B? Had we diligently followed Brown *et al*'s advice to avoid "pitfalls", made high-performing cell lines, and emphasized their preferred variable (technical precision) over the variable that was of more concern to us (variation within and between cell lines), we would in all likelihood have failed to perform the very measurement that ultimately led us to

the actual scientific findings of our original paper. Finally, although our original work was focused on the implications of Cas9-induced p53 activation for precision editing and for cell-based therapies, we would like to take this opportunity to also strongly advise against ignoring p53—the guardian of the genome—in any study that makes use of reagents that introduce double-strand DNA breaks.

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
