## [Review Process File · Molecular Systems Biology]

Response to the Correspondence „CRISPR Screens are feasible in TP53 wildtype cells” by Kevin Brown and colleagues

Emma Haapaniemi, Sandeep Botla, Bernhard Schmierer and Jussi Taipale

Review timeline:

Submission date:

17 June 2019

Accepted:

10 July 2019

Editor: Maria Polychronidou

Transaction Report:

1st Editorial Decision

10 July 2019

Thank you once again for sending us your Reply to the Correspondence by Moffat and colleagues. We have now received the comments of two reviewers who agreed to evaluate your Reply and who had also reviewed the Correspondence by Moffat and colleagues.

As you will see below, both reviewers think that the Reply is suitable for publication alongside the Correspondence by Moffat and colleagues. As such, we will now proceed with the publication of both pieces.

REFeree REPORTS

Reviewer #1:

I think that the comment by Taipale should be published, as it nicely explains and clarifies the different approaches the two groups took. However, I think that the comment should be published without the summary of "logical, technical and methodological flaws". I find it impossible to really referee the points brought up there and one would have to give the Moffat team the opportunity to respond to these points again. I do not think that this summary is necessary and the points are made well without it.

Reviewer #2:

We have been asked to review the 'Reply' from the Taipale lab to the correspondence manuscript from the Moffat lab (entitled 'CRISPR screens are feasible in TP53 wildtype cells'). Of note, we were one of the reviewers of the Brown manuscript and as a result of the review process, we now find the Brown manuscript suitable for publication in MSB.

The 'Reply' from the Taipale lab states that the main topic of the Brown manuscript is that the Taipale paper shows that CRISPR loss of function screening is not possible in p53 wildtype cells.

Here, I would disagree because the Brown paper is overall well balanced. Indeed, it explains that both original papers report important findings and indeed goes one step beyond, by explaining that key steps should be taken when reporting CRISPR findings. The Taipale 'Reply' agrees with these statements.

We also agree with the 'Reply' from the Taipale group where it is explained that the Moffat lab did not perform exactly the same experiment as the Taipale lab: the screening strategy was different, as was the library used etc. This was one of our comments on the Brown manuscript.

Therefore overall, I find the 'Reply' by Taipale to be acceptable for publication alongside the Brown paper. Neither make any incorrect statements and hence I think they should be circulated within the scientific community for their peers to assess.